# Computer Extracted Features from Initial H&E Tissue Biopsies Predict Disease Progression for Prostate Cancer Patients on Active Surveillance

**DOI:** 10.3390/cancers12092708

**Published:** 2020-09-21

**Authors:** Sacheth Chandramouli, Patrick Leo, George Lee, Robin Elliott, Christine Davis, Guangjing Zhu, Pingfu Fu, Jonathan I. Epstein, Robert Veltri, Anant Madabhushi

**Affiliations:** 1Department of Biomedical Engineering, Case Western Reserve University, Cleveland, OH 44106, USA; sxc868@case.edu (S.C.); pjl54@case.edu (P.L.); geoleemail@gmail.com (G.L.); 2Department of Anatomic Pathology, University Hospitals Cleveland Medical Center, Cleveland, OH 44106, USA; Robin.Elliott@uhhospitals.org; 3Department of Surgical Pathology, The Johns Hopkins Hospital, 1800 Orleans St, Baltimore, MD 21287, USA; cmd337@gmail.com (C.D.); gzhu6@jhu.edu (G.Z.); jepstein@jhmi.edu (J.I.E.); 4Department of Population and Quantitative Health Sciences, Case Western Reserve University, 10900 Euclid Ave, Cleveland, OH 44106, USA; pxf16@case.edu; 5Department of Urology and Oncology, The Johns Hopkins University, Baltimore, MD 21287, USA; rveltri11@comcast.net; 6Research Health Scientist, Louis Stokes Cleveland Veterans Administration Medical Center, 10701 East Blvd, Cleveland, OH 44106, USA

**Keywords:** prostate cancer, active surveillance, machine learning, pathology

## Abstract

**Simple Summary:**

Active surveillance (AS) prostate cancer patients suffer from a lower quality of life, increased risk of anxiety and depression, and an increased risk of disease progression compared to patients who opt for curative treatment. The current inclusion criteria for AS patients is unable to accurately identify patients with increased risk of progression, and therefore there is a need for a risk stratification technique that can identify patients with a higher risk of disease progression. In this work, we leverage quantitative histomorphometric (QH) features describing nuclear position, shape, orientation, and clustering from initial H&E biopsy images to accurately identify AS-eligible patients who are at high risk for disease progression. Our findings indicate that QH features were correlated with the risk of clinical progression in AS-eligible patients and was able to out-perform judgements based on clinical variables such as Gleason score and pro-PSA.

**Abstract:**

In this work, we assessed the ability of computerized features of nuclear morphology from diagnostic biopsy images to predict prostate cancer (CaP) progression in active surveillance (AS) patients. Improved risk characterization of AS patients could reduce over-testing of low-risk patients while directing high-risk patients to therapy. A total of 191 (125 progressors, 66 non-progressors) AS patients from a single site were identified using The Johns Hopkins University’s (JHU) AS-eligibility criteria. Progression was determined by pathologists at JHU. 30 progressors and 30 non-progressors were randomly selected to create the training cohort D_1_ (*n* = 60). The remaining patients comprised the validation cohort D_2_ (*n* = 131). Digitized Hematoxylin & Eosin (H&E) biopsies were annotated by a pathologist for CaP regions. Nuclei within the cancer regions were segmented using a watershed method and 216 nuclear features describing position, shape, orientation, and clustering were extracted. Six features associated with disease progression were identified using D_1_ and then used to train a machine learning classifier. The classifier was validated on D_2_. The classifier was further compared on a subset of D_2_ (*n* = 47) against pro-PSA, an isoform of prostate specific antigen (PSA) more linked with CaP, in predicting progression. Performance was evaluated with area under the curve (AUC). A combination of nuclear spatial arrangement, shape, and disorder features were associated with progression. The classifier using these features yielded an AUC of 0.75 in D_2_. On the 47 patient subset with pro-PSA measurements, the classifier yielded an AUC of 0.79 compared to an AUC of 0.42 for pro-PSA. Nuclear morphometric features from digitized H&E biopsies predicted progression in AS patients. This may be useful for identifying AS-eligible patients who could benefit from immediate curative therapy. However, additional multi-site validation is needed.

## 1. Introduction

For low grade prostate cancer (CaP) patients, active surveillance (AS) is an increasingly favored management strategy that involves repeat blood work, biopsies, and digital rectal exams [1]. However, AS patients express a significant decrease in their quality of life and an increased risk of anxiety and depression [1]. Repeat biopsies also increases the risk of infection [2]. Delaying curative treatment also introduces added complications of disease progression and metastasis [1]. As high as 43% of AS patients eventually opt for curative therapy due to biopsy progression or anxiety [3]. Physicians use many clinical risk factors to balance the risk of CaP progression and overtreatment.

These risk factors include prostate specific antigen (PSA) density, clinical stage, number of positive cores, and biopsy Gleason grade. Various institutions have defined different thresholds for these risk factors to define AS-eligibility criteria [4,5,6,7]. However, because no perfect threshold exists, these limited number of clinical factors are unable to accurately ascertain the likelihood that a patient will eventually need curative intervention [8,9]. Depending on different combinations of thresholds, between 12% to 31% of AS-eligible demonstrate clinical levels of progression and require curative treatment [8]. The difficulty in identifying patients for whom AS is appropriate is further compounded by the low specificity of PSA elevation [4,10,11], and low inter-reviewer agreement in Gleason grading [11].

Recently, there has been an increased interest in determining, within a cohort of AS patients, features that may be associated with a greater risk of clinically significant progression. Several groups identified serum, urinary, and tissue biomarkers linked to aggressive CaP morphology in AS-eligible patients [12]. However, these markers have yet to be directly correlated with patients who progress or are unable to determine the risk when the patient is initially placed in AS [4,13,14,15,16]. Therefore, there is a need for an accurate, quantifiable method that can determine early whether a patient is at an increased risk of significant progression in CaP.

With the increase in computing power, features extracted from initial biopsies can directly characterize the tumor morphology and predict outcome instantly and accurately. Quantitative histomorphometric (QH) features are computationally derived descriptors of morphology extracted from digitized pathology slides. QH approaches have shown to be useful in characterizing CaP in a number of studies [17,18,19,20,21,22], including through the use of deep learning, in which features are automatically generated via a cascade of neural networks [23,24]. While QH features have been employed for CaP detection, grading, and biochemical recurrence prediction, no previous study, to our knowledge, has attempted to associate QH features with the risk of clinical progression in AS patients.

In this study, we explore the potential role of QH features including spatial arrangement, shape, and disorder of nuclei from within cancer defined regions on initial H&E biopsies to differentiate AS patients who are at low and high risk of clinical progression. A total of *n* = 191 AS patients were identified from a single institution with a corresponding initial Hematoxylin & Eosin (H&E) biopsy. Supervised machine learning classification approaches were employed to evaluate the ability of nuclear QH features extracted from digitized H&E images of baseline biopsies in predicting risk of progression. The ability of these QH features to predict likelihood of progression was also further compared against a known CaP biomarker, pro-PSA, in *n* = 47 patients. Finally, the QH model was compared to a deep learning network-based model.

## 2. Materials and Methods

### 2.1. Dataset

Digitized H&E needle-core biopsy images from 191 AS patients (125 progressors, 66 non-progressors) from Johns Hopkins University (JHU) who met the criteria of stage T1c, PSA ≤ 10 ng/mL, Gleason sum (GS) ≤ 7, ≤ 2 positive cores, ≤ 50% core involvement, PSA density ≤ 0.15 ng/mL and life expectancy ≤ 20 years were gathered. All cases used for TMAs and AS biopsies were consented under an Institutional Review Board-approved protocol at Johns Hopkins University School of Medicine. De-identified images with no patient header information (PHI) was provided to the CWRU team and hence from an analysis perspective the research we deemed to be human subjects exempt. Histopathologic confirmation of upgrading and upstaging was determined by pathologists at JHU to differentiate progressors and non-progressors. Specimens were digitized at 40× magnification (0.25 microns per pixel). All studies were conducted in a singled representative CaP region per patient annotated by a pathologist. 30 progressors and 30 non-progressors were randomly selected to constitute the training set D1 (*n* = 60) and the remaining set of 131 (95 progressors, 36 non-progressors) patients comprised the test set D2. A subset (D3, *n* = 47) of D2 also had pro-PSA measurements.

### 2.2. Nuclear Segmentation

Each H&E stained image was first normalized to the same color intensity level by adjusting the color levels of the image using the maximum stain vector as a template [25]. Nuclei were detected by convoluting a bank of directional Gaussian filter kernels with the normalized H&E images to generate a response map [24]. 

Areas of local maximums in the response map indicated high probability of a nucleus. Each nucleus was then automatically segmented by a watershed-based nuclear segmentation method [25]. From these segmentations, the boundary points and centroid of every nucleus was extracted. An example segmentation result is shown in Figure 1.

### 2.3. Feature Extraction

Subsequent to nuclear segmentation, a total of 216 nuclear morphology features were extracted from the segmentation results. These features belonged to four categories: graph, shape, nuclear disorder, and cluster graph.

Graph features were constructed based on nuclear centroids. Voronoi tessellation, Delaunay triangulation, minimum spanning trees, and k-nearest neighbor plots [17,18] were constructed over the entire annotated region of each image. Statistics such as mean and standard deviation were derived from the areas and perimeters of Voronoi polygons, Delaunay maps, minimum spanning tree edge lengths, and distances to nearest 3, 5, and 7 nuclear centroids [17,18].

The nuclear shape-based features were derived from the boundary of the segmented nuclei. These measures included nuclear area, perimeter, smoothness, fractal dimension, and invariant moments, and Fourier descriptors [19,20,21,22]. For each measure, the mean, median, standard deviation, and minimum to maximum ratio statistics were extracted.

The nuclear disorder features measure the disorder in the orientation of the nuclei within a local neighborhood [18,26]. The orientation of each nucleus was determined using the first principal component of the boundary points. This orientation vector roughly corresponds to the direction of the major axis of the nucleus. Local subgraphs of the nuclear arrangement were created with a probabilistic decay model based on Euclidean distance and an empirically determined decaying factor [18]. A co-occurrence matrix was created using the relative orientation of each nucleus in each subgraph. Measures of entropy, energy, and contrast were extracted from this matrix [18].

The cell cluster graph features use clusters of nuclei as nodes and capture the local and global spatial relationships between each cluster [27,28]. These spatial relationships include edge length, connectivity coefficients, and clustering coefficients. These features are shown in Figure 1 and are summarized in Table 1.

### 2.4. Feature Identification

In order to identify the optimal model and feature set, the 216 features from Table 1 were evaluated on D_1_ for stability and accuracy. D_1_ was randomly divided into two cohorts, cohort D_1,A_ and cohort D_1,B_, each with 30 patients (15 progressors, 15 non-progressors). Discriminating features were first identified on cohort D_1,A_. Two sets of six discriminating feature were selected, one by Wilcoxon rank-sum (WLCX), the other by *t*-test (TT). Each of these sets were then used to train two classifiers on cohort D_1,A_, one using linear discriminant analysis (LDA), the other using quadratic discriminant analysis (QDA). The patients in cohort D_1,B_ were then classified using each of these four models (TT + QDA, TT + LDA, WLCX + QDA, WLCX + LDA). The AUC of each model on cohort D_1,B_ was recorded. Cohorts D_1,A_ and D_1,B_ were then randomly redrawn from the training set and the feature identification and classifier training process was repeated 30 times.

From the 30 iterations, only feature combinations yielding an AUC ≥ 0.65 were retained. This threshold was chosen empirically to maximize predictive potential and minimize overfitting. The six most recurring features among the retained feature combinations were deemed discriminatory and used for classifier construction. These six most recurring features are listed in Table 2.

Hierarchical clustering of the top six features in D_1_ was performed using spearman distance to further evaluate the top six features’ ability in differentiating outcome [28]. Clustering results were visualized on D_1_ using a heatmap where green indicates higher expression of the feature values and brown indicates the converse. These unsupervised clustering results were compared against patient outcome to evaluate associations between feature expression and patient outcome. 

### 2.5. Classifier Construction

The 6 features identified in the previous step were used to create three models for progression prediction. These models were constructed using LDA, QDA, and random forest (RF) classifiers. 3-fold cross validation was used in 100 iterations to identify the model with the highest average AUC. The optimal operating point for the best model was determined on D_1_. The best model and the operating point was then evaluated on D_2_ to determine AUC and accuracy. A confusion matrix was constructed using the best model and the operating point on D_2_ and displayed in Table 3. To establish a comparison point for QH features, the AUC of a two clinical predictors (pro-PSA, and biopsy Gleason sum) was investigated.

### 2.6. Deep Learning Model

A deep learning model using the DenseNet architecture [29] was trained to compare to the QH models. The deep learning model takes images as input, splits them into 224 × 224 pixel patches at 1 MPP (equivalent to 10× magnification), and renders a progressor/non-progressor decision on each patch. Model hyperparameters were chosen to produce a smaller version of the DenseNet121 model more suitable for a small dataset, resulting in a model with a 64 features in the first layer and an additional 12 in each layer with a feature multiplier of 4 in the bottleneck layers, two layers in each pooling block, and no dropout, for a total of 79,784 parameters. This model was trained on D1, split into a 50 patient training set and 10 patient testing set, for 500 epochs, with the final model being the one with the lowest loss on the test set. This model was then applied to D2, on which per-image AUC, based on fraction of positive patches, and accuracy, based on the majority vote of the patches, were calculated.

## 3. Results

### 3.1. Experiment 1: Identifying Features Associated with Risk of Progression in AS Patients

The top six nuclear QH features obtained during feature identification within D_1_ are listed in Table 2. Three nuclear shape features were represented in the six feature set, two nuclear spatial arrangement features, and a single feature representing the localized nuclear disorder.

As can be seen in Figure 2, there exists good separation between patient outcome and feature values. For all the shape features (columns 1, 2, 4), an increasing feature value is associated with a better prognosis. For the Voronoi polygon area feature (column 6), an increasing feature value is closely associated with clinical progression. Similarly, a high nuclear disorder feature (column 5) seems to be strongly associated with an increased risk of clinical progression. A high Voronoi polygon perimeter feature (column 3) value appears to associate with a better disease prognosis.

### 3.2. Experiment 2: QH based Classifier Construction to Predict Progression in AS Patients

Results from the three-fold cross validation with LDA, QDA, and RF models using the top six features are shown in Table 4. The RF model yielded an AUC of 0.75 in D_2_. Within D3, the RF model yielded an AUC of 0.79, surpassing both pro-PSA (AUC = 0.42) and Gleason (AUC = 0.64).

### 3.3. Experiment 3: Deep Learning Classifier for Prediction of Progression in AS Patients

The deep learning model had an AUC of 0.64 based on fraction of positive patches and accuracy of 0.73 based on patch majority voting in D_2_, performing worse than the random forest model. While the deep learning model had a specificity of 0.96, its sensitivity was just 0.14. This bias arose despite the deep learning model’s training set consisting of an equal number of progressor and non-progressor cases.

## 4. Discussion

Patient inclusion criteria for prostate cancer (CaP) patients on active surveillance is not an exact science. Many studies have suggested that the current inclusion criteria fails to accurately exclude patients at a high-risk of CaP progression [1,8,9]. Therefore, there has been an increased effort to determine factors correlated with an increased risk of progression in AS patients. Multivariate nomograms have been tested with AS patients to track CaP progression, however average AUCs reported were 0.59 [30]. Some groups have also attempted to identify serum, and urinary clinical biomarkers to determine the risk of progression in disease using protein expression, and tumor morphology [12,15,16]. For example, PCA3 and pro-PSA have been studied as predictors of more aggressive disease. However, PCA3 possesses low predictive power (AUC = 0.66) and provides no increased independent predictive value compared to Gleason score [4,12]. Pro-PSA has been reported to predict histologic upgrading and upstaging with a c-index of 0.62 for AS-eligible patients, however lacks a comparative analysis to other clinical variables such as Gleason score [4,31].

In this work, we evaluated the ability of QH features relating to nuclear spatial arrangement, shape, and disorder to quantitatively describe tumor morphology from an initial H&E needle core biopsy and predict the likelihood of progression in AS patients. We validated this model on an independent validation set (*n* = 131) and compared it against pro-PSA, a form of PSA that is more linked with CaP, and Gleason sum on a subset (*n* = 47) using AUC. Additionally, we compared our model to a deep learning-based method.

Several studies have evaluated the use of QH features to characterize nuclear and glandular morphology for CaP detection, Gleason grading, and BCR prediction. Doyle et al. used 103 features describing nuclear and glandular spatial arrangement, and texture to classify H&E biopsies into Gleason grade 3 and 4 with an accuracy of 76.9%. Other studies have exceeded the limitations of Gleason grade by relating the nuclear and glandular architecture directly with patient outcome. Lee et al. derived 39 features describing entropy in nuclear orientation from CaP tissue microarray (TMA) to predict biochemical recurrence post-radical prostatectomy (RP) [26]. The localized cell clustering has been used to predict 5-year BCR using TMAs with an accuracy of 83.1% [32].

To our knowledge, the work presented in this study, represents the first attempt to use computerized histologic image analysis and machine learning approaches to predict likelihood of CaP progression in AS patients based off routine H&E biopsy tissue images.

We found that the smaller the variance in the roundness of the nuclear perimeter, the better the prognosis. This suggests that a greater degree of nuclear shape homogeneity is associated with a lower risk of clinical progression, a finding consistent with prior work in CaP and breast cancer [11,18,19,32,33,34,35]. The Voronoi polygons constructed using the nuclear centroids showed a relationship between density of nuclear packing and patient outcome. Patients that exhibited greater variability in density of nuclear packing on the initial biopsy were associated with an increased likelihood of clinically significant CaP progression, a conclusion that is consistent with other studies that found decreasing organization of nuclear arrangement closely related to more aggressive disease [36]. Finally, the nuclear orientation feature is a second order nuclear angle statistic representing the range of the orientations of the nuclei in local subgraphs on an image. In this dataset, a smaller range of nuclear orientations present on a biopsy slide was associated with a better prognosis. Lee et al. has reported similar results with glandular [18] and nuclear arrangements in predicting BCR [26].

The combination of the six QH features yielded a higher predictive power (AUC = 0.75) than the average reported predictive capability of standard nomograms (average AUC = 0.59) [30]. Due to the low specificity of PSA, groups have investigated the use of pro-PSA, a variant of PSA, in differentiating benign and aggressive disease [31,37,38,39,40]. While pro-PSA has been reported to be significantly correlated with upgrade in disease, in this dataset the pro-PSA had very low discriminating ability (AUC = 0.42) when compared to the QH model (AUC = 0.79) [31,38].

The deep learning method (AUC = 0.64) performed worse than the QH model, with a sensitivity (0.14) far too low to be clinically useful. While it is possible that further refinement of the model could improve these results, it is clear that deep learning does not provide a simple solution to prediction of progression from AS.

Our study did however have its limitations. First, in the 191 patient dataset, only 47 patients in our study had pro-PSA information for us to compare to. Second, we acknowledge that one of the limitations of our study was the lack of detailed manual annotations of individual nuclei that precluded us from comprehensively evaluating the sensitivity of the feature extraction process on fidelity of the nuclear contours. Our dataset also lacked time-to-progression information required to conduct a survival analysis of our model. Our dataset was missing clinical information needed to compare the performance of our QH model with that of nomograms. Our entire process was only conducted in a small region of interest identified by a pathologist rather than the entire whole slide image of the biopsy specimen. Our study was also conducted using patients from a single institution. Finally, while we did not explicitly address the issue of tumor heterogeneity, it was interesting that the QH features from a single tumor focus was associated with likelihood of progression. A direction of future work would be to explicitly invoke multiple different tumor foci and evaluate whether the associated QH measurements could even better prognosticate likelihood of disease progression. Future work will also need to involve a larger, multi-site, independent validation of these findings.

## 5. Conclusions

In this study, we demonstrated the ability of QH features extracted from initial needle core biopsies to identify AS-eligible patients at high risk of progression. While our QH model was found to be accurate on a separate test set and appeared to outperform a clinically employed CaP biomarker, additional multi-site testing is needed prior to use of this approach for clinical management of CaP patients on AS.

The deep learning method (AUC = 0.64) performed worse than the QH model, with a sensitivity (0.14) far too low to be clinically useful. While it is possible that further refinement of the model could improve these results, it is clear that deep learning does not provide a simple solution to prediction of progression from AS based off digital biopsy scans. Furthermore, there is no straightforward way to understand the rationale for the decisions of the deep learning model, obfuscating model interpretability. It may be that much larger training sets might be needed to develop more robust deep learning based prognostic models for prostate cancer in the active surveillance setting.

Utilizing QH features to aid in selecting ideal candidates for AS could decrease the rate of overtreatment in low-risk CaP patients with minimal additional costs as it only requires digitization of an initial biopsy and a standard desktop computer. While digital pathology services are not currently routinely performed in every pathology lab, with the recent Food and Drug Administration (FDA) approval of whole slide scanners from two different vendors, it is highly likely that slide digitization will become more routine and prevalent in the coming years [41].

## Figures and Tables

**Figure 1 cancers-12-02708-f001:**
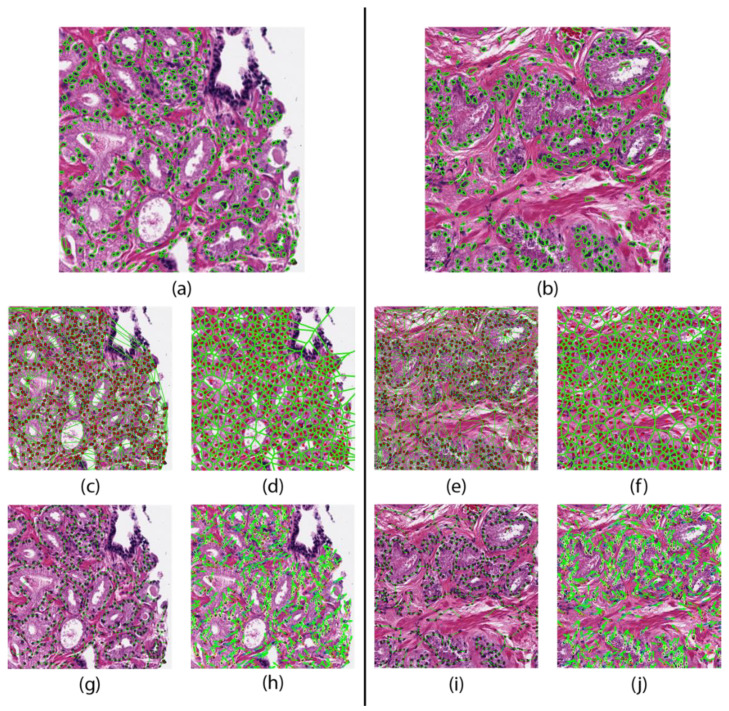
Feature maps of a patient that did not progresses (left) and one who progressed (right). Nuclear segmentation results representing the nuclear shape (**a**,**b**). Nuclear spatial arrangement represented through the Delaunay triangulation (**c**,**e**) Voronoi tessellation (**d**,**f**), and Minimum Spanning Trees (**g**,**i**). Disorder in localized nuclear orientation (**h**,**j**).

**Figure 2 cancers-12-02708-f002:**
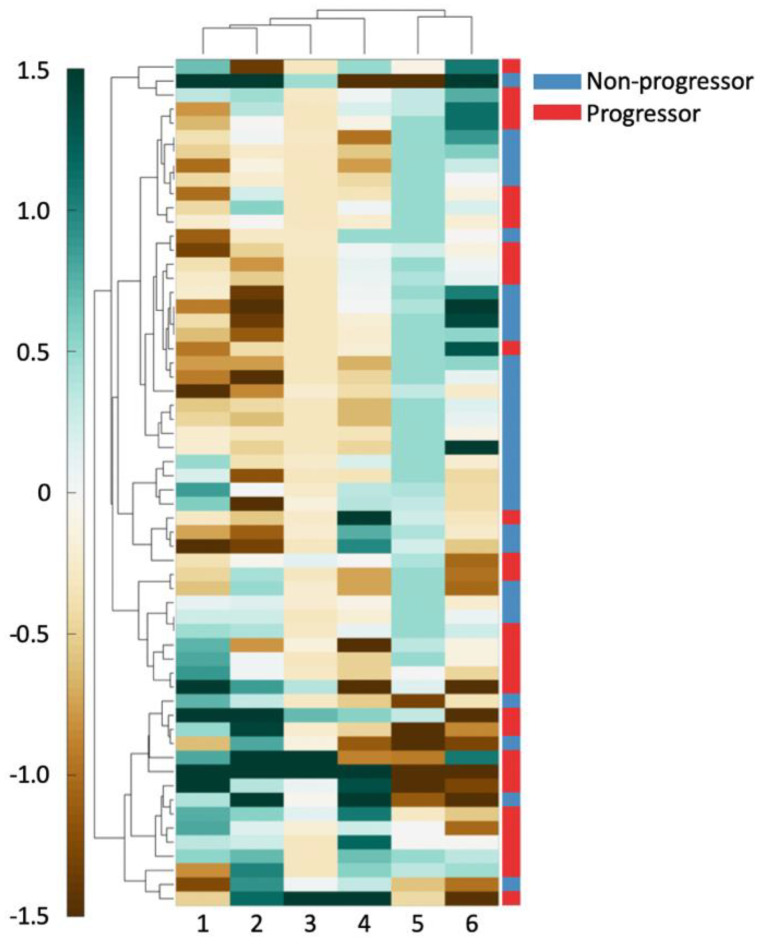
The 6 features used to create a clustergram on D_1_. Features and patients were clustered hierarchically. The shading of each cell shows the relative over- or under-expression of that feature in a patient. From left to right, the features are: (1) standard deviation of minimum/maximum nuclear radius, (2) minimum/maximum nuclear perimeter, (3) minimum/maximum ratio of Voronoi polygon perimeter, (4) standard deviation of nuclear shape’s Fourier descriptor 6, (5) tensor contrast inverse moment range, and (6) standard deviation of Voronoi polygon area.

**Table 1 cancers-12-02708-t001:** Summary of QH based features to describe tumor morphology.

Feature Family	Description	Features
Graph	Voronoi, Delaunay, minimum spanning trees, k-NN graphs	51
Shape	Area, perimeter ratio, smoothness, distance, etc.	100
Nuclear Disorder	Orientation entropy, energy, contrast	39
Cluster Graphs	Clustering coefficient, edge length, connected components	26

**Table 2 cancers-12-02708-t002:** The six features identified from D_1_ and used in the final model.

Feature
Voronoi: Min/max polygon perimeter
Shape: Min/max standard deviation of distance of contour point from centroid of nuclei
Shape: Min/max nuclei perimeter
Shape: Standard deviation of Fourier descriptor 6
Orientation: Tensor contrast inverse moment range
Voronoi: Standard deviation of polygon area

**Table 3 cancers-12-02708-t003:** A confusion matrix created on D2 using the RF model using an operating point threshold of 0.69. The positive predictive value (PPV), negative predictive value (NPV), sensitivity and specificity was also calculated on D2 using the same operating point.

		Predicted		
**Actual**		Non-Progressor	Progressor	
	Non-Progressor	22	14	NPV: 44%
	Progressor	28	67	PPV: 83%
		Specificity: 61%	Sensitivity: 71%	

**Table 4 cancers-12-02708-t004:** The average AUC, accuracy (ACC), sensitivity (SENS), and specificity (SPEC) across 100 iterations for each classification scheme on D_1_ using the 6 discriminating features from Experiment 1. These values were calculated using 3-fold cross validation. Each fold had an equal number of progressors (*n* = 10) and non-progressors (*n* = 10).

Model Type	AUC	ACC	SENS	SPEC
LDA	0.68 ± 0.12	0.63 ± 0.11	0.53 ± 0.16	0.26 ± 0.13
QDA	0.69 ± 0.11	0.62 ± 0.10	0.58 ± 0.17	0.67 ± 0.16
RF	0.73 ± 0.10	0.75 ± 0.09	0.70 ± 0.17	0.81 ± 0.16

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
