# Peer review of "Computer Extracted Features from Initial H&E Tissue Biopsies Predict Disease Progression for Prostate Cancer Patients on Active Surveillance"

_cancers, 2020, doi:10.3390/cancers12092708_

Round 1

Reviewer 1 Report

There appears to be some missing references.

it is unclear to me as to how progression is defined.  Was this on clinical grounds, or a rise in psa, or routine biopsy upgrading, for example?

I was unclear as to how many pathology labs could do this digital pathology to arrive at a prediction of future progression.

It would be useful if H&E slides could be interrogated digitally to provide information over and above Gleason grade, which would be 6 in most of these active surveillance patients.

As a proof of concept it is a step in the right direction.  However, I suspect a genetic analysis of the tumour biopsy might prove more useful in the long run.

Reviewer 2 Report

The authors present a study in which conventional feature engineering and supervised classification (random forest) approaches are applied to a set of digital images from prostate adenocarcinoma. These features focus on nuclear morphometry and spatial organization. These features were then used to attempt to predict progression in the context of active surveillance in prostate cancer.

Major Critiques

First, the authors should consider a more complete consideration of the clinical context and therefore the implication for the model tuning. In the case of active surveillance this is not a "balanced" loss matrix. I would argue that a false positive is a far more significant error in this context as compared to a false negative because a false positive would result in a radical prostatectomy that is not medically warranted, while a false negative would be addressed by subsequent biopsy on active surveillance. I would ask the authors to consider embedding this logic in the feature selection and model training process by considering other metrics than just AUC that could potentially select features and subsequent model that can be more specific even if less sensitive. One might even consider addressing this with the models that were already developed in the manuscript simply by adjusting the thresholds in the training data to achieve higher specificity (although I would think that starting from the full set of features would improve performance). 

Second, it is not clear to me why the D1 training dataset was restricted to only 60 samples and then the D2 hold-out validation had 131 (with the D3 subset of D2 have the 47 sample that had complete clinical correlates for comparisons). Generally speaking, more data is better in machine learning (assume the data is all of the same quality). Therefore at first glance I would seriously consider running these analyses again with a training cohort of ~ 100 samples and then have 91 left for the hold-out validation. That could even be pushed up more arguably ~120 samples for training, since the most important samples for validation (the 47 D3 cohort) are really the key samples that you want to hold-out for validation.

Minor Comments

Its hard to image not considering an alternative to feature engineering + random forests, namely the application of deep learning techniques to these data. One may consider at least mentioning that in the discussion. The approach in the manuscript is dependent on using domain knowledge to extract features - and it very well might be the case that better features exist outside that set.

The labels in Figure 1 appear to be repeated (c) and (e)

The color palettes in Figure 2 need to be changed as they are very hard to interpret. Would suggest the conventional blue-white-red for the heatmap and then may green and yellow for the labels of no progression vs progression. Additionally the shape of this heatmap would be far better if each cell were not as wide.

Round 2

Reviewer 1 Report

An interesting concept which will help better stratify patients for active surveillance, if able to be applied in clinical /pathology practice.

Reviewer 2 Report

I appreciate the effort the authors have taken to thoughtfully respond to my comments.

The deep learning approach presented in the manuscript is valid and offers a comparison to the feature engineering approach; however, it's a bit over-simplified and unlikely to be optimized in particular in regards to how features from tiles are aggregated over a whole image. That being said, the authors do accept that this implementation can be improved upon in the discussion. Therefore, overall I find the authors' response and changes here acceptable.

The confusion matrix presented is both helpful and informative but does need to be adjusted to more clearly delineate which is ground truth vs prediction labels (rows vs columns) and I would think positive/negative predictive values and/or sensitivity/specificity should be included to more fully present these data to the reader. 
